

# Demographic inference through approximate-Bayesian-computation skyline plots

Miguel Navascués[1,2], Raphaël Leblois[1,2] and Concetta Burgarella[3]

[1] CBGP, INRA, CIRAD, IRD, Montpellier SupAgro, University of Montpellier, Montpellier, France
[2] Institut de Biologie Computationnelle, Montpellier, France
[3] UMR DIADE, IRD, Montpellier, France

## ABSTRACT

The skyline plot is a graphical representation of historical effective population sizes as a function of time. Past population sizes for these plots are estimated from genetic data, without *a priori* assumptions on the mathematical function defining the shape of the demographic trajectory. Because of this flexibility in shape, skyline plots can, in principle, provide realistic descriptions of the complex demographic scenarios that occur in natural populations. Currently, demographic estimates needed for skyline plots are estimated using coalescent samplers or a composite likelihood approach. Here, we provide a way to estimate historical effective population sizes using an Approximate Bayesian Computation (ABC) framework. We assess its performance using simulated and actual microsatellite datasets. Our method correctly retrieves the signal of contracting, constant and expanding populations, although the graphical shape of the plot is not always an accurate representation of the true demographic trajectory, particularly for recent changes in size and contracting populations. Because of the flexibility of ABC, similar approaches can be extended to other types of data, to multiple populations, or to other parameters that can change through time, such as the migration rate.

## INTRODUCTION

Inferring the historical demography of populations by means of genetic data is key to many studies addressing the ecological and evolutionary dynamics of natural populations. Population genetics inference, with appropriate dating, can identify the likely factors (such as climatic events) determining the demography of a species. With enough research resources, this can be done with outstanding detail (e.g., in humans, reviewed in *Nielsen et al., 2017*). Demographic inference can also be used to generate null models for the detection of loci under selection (as discussed in *Hoban et al., 2016*).

At present, most of the methods to estimate demography from genetic data are based on the coalescent. The coalescent (see *Wakeley, 2008*, for a review) is a mathematical model that describes the rate at which genetic lineages coalesce (i.e., join in a common ancestor)

Corresponding author
Miguel Navascués,
miguel.navascues@inra.fr

towards the past, forming the genealogy of the sample. The coalescence probability depends on the effective population size at each time in the past; that is, the demographic history of the population. Given a genealogy, the coalescent enables a calculation of the likelihood of the demographic model. Demographic inference is obtained by calculating the likelihood of the model given the data, which requires integrating over all possible genealogies for the data. This is approximated by means of Monte Carlo algorithms known as coalescent samplers (see review by *Kuhner, 2009*).

Alternatively, the coalescent can be used to calculate the likelihood of the number of genetic differences for a pair of gene copies under a given demographic model. The likelihood for all pairs in a sample can be combined to obtain a composite-likelihood (which is not a true likelihood because pairs are not independent and they are related by their genealogy). The composite-likelihood score can be used as a criterion to estimate the parameters of the model with faster algorithms than the coalescent samplers although with lower performance, particularly regarding confidence intervals (e.g., *Navascués, Hardy & Burgarella, 2009*; *Nikolic & Chevalet, 2014*).

Coalescent models can also be used in the likelihood-free framework known as Approximate Bayesian Computation (ABC, *Tavaré et al., 1997*; *Beaumont, Zhang & Balding, 2002*). In this approach, the likelihood is substituted by the similarity between the observed data and simulated data generated from a given model. Similarity is usually evaluated by means of a distance between observed and simulated summary statistics. This distance allows one to select the simulations close to the observed data and reject those too far away. Posterior probability distributions are estimated from the collection of parameter values used in the selected simulations (see *Beaumont, 2010* for a review on ABC).

A classical way to address the estimation of past population size changes by these methods is to assume simple parametric models, such as exponential, logistic or instantaneous demographic change. However, these are sometimes considered too simple to describe the dynamics of real populations. In the skyline plot methods, the underlying demographic model is a piecewise constant population size model, i.e., the demographic history consists of several periods of constant size, with instantaneous changes of sizes between consecutive periods. The aim of this model is to provide a more flexible framework that could capture the complex demography expected in natural populations. Skyline plots were introduced by *Pybus, Rambaut & Harvey (2000)* who estimated the effective population size in the time intervals defined by the coalescent events of a given genealogy (which was considered as known) from the expected waiting time between coalescent events. The graphical representation of those estimates suggests the skyline of a city, giving the name to the method. Such models have been implemented in a Markov chain Monte Carlo coalescent sampler (BEAST software; *Drummond et al., 2005*; *Minin, Bloomquist & Suchard, 2008*; *Heled & Drummond, 2008*), and in an importance sampling coalescent sampler (*Ait Kaci Azzou, Larribe & Froda, 2015*) for the analysis of sequence data. The addition of microsatellite mutation models to BEAST (*Wu, Drummond & Uyenoyama 2011*) made it possible to infer skyline plots from this type of data (e.g., *Allen et al., 2012*; *Molfetti et al., 2013*; *Minhós et al., 2016*). Also for microsatellite data, a composite-likelihood approach has been developed  (R package VarEff, *Nikolic & Chevalet, 2014*).

Similar piecewise models to infer historical population sizes through time have been proposed in the context of population genomics (e.g., *Li & Durbin, 2011*; *Terhorst, Kamm & Song, 2016*). The methods discussed above assume a set of independent (unlinked) genetic markers. However, if a large proportion of the genome has been sequenced, the studied polymorphism are in linkage disequilibrium. Methods such as the Pairwise Sequentially Markovian Coalescent (PSMC, *Li & Durbin, 2011*) and its successors profit from the additional information of linkage disequilibrium for the inference. We will not further discuss this family of methods, as our focus here is on datasets of independent molecular markers, such as microsatellites, which remain reliable markers for low-budget projects. Note, however, the PSMC-like implementation on ABC by *Boitard et al. (2016)*.

The use of the skyline plot in the ABC framework was first proposed in *Burgarella et al. (2012)*. Here, we provide a suite of R scripts (DIYABCskylineplot) to produce approximate-Bayesian-computation skyline plots from microsatellite data and evaluate its performance on simulated pseudo-data. We show the method to be useful for detecting population decline and expansion and discuss its limits. ABC skyline plots are then built for four study cases (whale shark, leatherback turtle, Western black-and-white colobus and Temminck's red colobus) and compared with the demographic inference obtained by an alternative full likelihood method.

## METHODS

### ABC skyline plot

For a demographic skyline plot analysis within the ABC framework, our model consisted of a single population with constant size that instantaneously changes to a new size $n$ times through time. The parameters (from present to past, as in the coalescent model) are the present scaled population size $\theta_0 = 4N_0\mu$ (where $N_0$ is the effective population size in number of diploid individuals and $\mu$ is the mutation rate per generation) which changes to $\theta_1$ at time $\tau_1 = T_1\mu$ (where $T$ is the time measured in generations), remains at $\theta_1$ and then it changes to $\theta_2$ at $\tau_2$, and so on, until the last change to $\theta_n$ at $\tau_n$. Note that other models and parametrization could have been used for our purpose, as in the alternative model that we present in Section S1.2.

The objective of a standard ABC analysis would be to estimate the posterior distribution for each parameter of the model. In our case, the parameters $\{(\theta_i, \tau_i); i \in [0, n]\}$ have been treated as nuisance parameters and we focused on inferring from them the trajectory of the scaled effective population size along time, $\theta(t)$, as in *Drummond et al. (2005)*. In order to approximate $\theta(t)$ we select $m$ times of interest, $\{t_j; j \in [1, m]\}$. Given a simulation $k$ with parameters $\{(\theta_{k,i}, \tau_{k,i}); i \in [0, n_k]\}$, derived parameters $\{\theta_k(t_j); j \in [1, m]\}$ are obtained as follows: $\theta_k(t_j) = \theta_{k,i}$ for $i$ satisfying the condition $\tau_{k,i} \leq t_j < \tau_{k,i+1}$ (see Fig. S1 for some examples). For each $t_j$, inference of the derived parameters $\theta(t_j)$ were obtained following standard ABC procedures as described elsewhere (e.g., *Beaumont, Zhang & Balding, 2002*). Median and 95% highest posterior density (HPD) intervals of derived parameters $\theta(t_j)$ were used to draw ABC skyline plots.

Simulations with different numbers of population size changes can be used for inference because of the use of derived parameters $\theta(t_j)$, which are common to all models. We set

the prior probability on the number of constant size periods to be Poisson distributed with $\lambda = \ln(2)$ as in *Heled & Drummond (2008)*. This gives equal prior probability to stable populations (a single period of constant size) and changing populations (two or more periods). Thus, posterior probability on the number of periods may be used to discriminate between stable and changing demographies by estimating the Bayes factor of one period (constant population size) *versus* several demographic periods (variable population size). Posterior probabilities of contrasting models can be obtained by logistic regression as described elsewhere (*Beaumont, 2008*).

We implemented this approach in a suite of R scripts (*R Core Team, 2017*) that we named DIYABCskylineplot (*Navascués, 2017*). For each simulation the number of population size changes is sampled using the prior probabilities. Through a command line version of DIYABC (v2.0, *Cornuet et al., 2014*), parameter values $\{(\theta_{k,i}, \tau_{k,i}); i \in [0, n_k]\}$ are sampled from the prior distribution, coalescent simulations are performed and summary statistics are calculated (mean across loci of the number of alleles, $N_a$; heterozygosity, $H_e$; variance of allele size, $V_a$, and *Garza & Williamson (2001)* statistic, $M$). In addition, the *Bottleneck* statistic ($\Delta H$; *Cornuet & Luikart, 1996*), which compares the expected heterozygosity given the allele frequencies with the expected heterozygosity given the observed number of alleles, is calculated in R from the summary statistics provided by DIYABC. Derived parameter values, $\{\theta_k(t_j); j \in [1, m]\}$, are calculated from the reference table (i.e., table of original parameters and summary statistics values for all simulations) produced by DIYABC and their posterior probability distributions are estimated in R using the *abc* package (*Csilléry, François & Blum, 2012*).

## Simulations

The method described above was evaluated on simulated data (pseudo observed data-set, POD) of contracting and expanding populations. Declining populations had a present effective size of $N_0 = 100$ diploid individuals that changed exponentially until time $T$, which had a value of 10, 50, 100 or 500 generations in the past, reaching an ancestral population sizes of $N_A$, which had a value of 1,000, 10,000 or 100,000 individuals. Expanding populations had a present population size of $N_0$ with a value of 1,000, 10,000 or 100,000 diploid individuals, which changed exponentially until reaching the size of the ancestral population $N_A = 100$ at time $T$, which had a value of 10, 50, 100 or 500 generations in the past. For times older than $T$, the population size is constant at $N_A$ for all scenarios. In addition, we simulated three constant population size scenarios with $N$ taking a value of 1,000, 10,000 or 100,000. Equivalent scenarios were also evaluated in *Girod et al. (2011)* and *Leblois et al. (2014)*. PODs were generated for 50 individuals genotyped at 30 microsatellite loci evolving under a generalised stepwise mutation model (GSM, *Slatkin, 1995*). Additional PODs varying in number of loci (7, 15 or 60 loci) and sample size (6, 12, 25 or 100 diploid individuals) were produced to evaluate the influence of the amount of data in the detection of demographic change. Mutation rate was set to $\mu = 10^{-3}$ and $P_{GSM}$ to 0, 0.22 or 0.74 ($P_{GSM}$ is the parameter of a geometric distribution determining the mutation size in number of repeats). One hundred replicates (i.e., PODs) were run for each scenario. Therefore, the mutation scaled parameter values are for $\theta = 4N\mu$: 0.4, 4,

40 or 400 and for $\tau = T\mu$: 0.01, 0.05, 0.1 or 0.5. PODs were obtained using the coalescent simulator fastsimcoal (*Excoffier & Foll, 2011*).

Every POD was analysed with the same set of prior probability distributions that largely includes all parameter values of simulations. Scaled effective size parameters, $\theta_i$, were taken from a log-uniform distribution in the range $(10^{-3}, 10^4)$ and scaled times, $\tau_i$, from a log-uniform distribution in the range $(2.5 \times 10^{-4}, 4)$. A uniform prior in the range $(0, 1)$ was used for mutational parameter $P_{GSM}$. For each replicate of each scenario, we obtained the skyline plot (median and 95% HPD intervals of the $\theta(t_j)$ posterior distributions) and estimated the Bayes factor between constant size and variable demography by using logistic regression. Estimates of the mutational parameter $P_{GSM}$ were also obtained for each POD. For each scenario, mean absolute error, bias and proportion of times the true value falls outside the credibility interval were estimated.

## Data sets

In addition to PODs, four real data-sets from the literature were re-analysed with the ABC skyline plot described above. The first data-set comes from the whale shark (*Rhincodon typus*), the largest extant fish. Whale sharks inhabit all tropical and warm temperate seas and are considered an endangered species with a global population decline of more than 50% in the last three generations (*Pierce & Norman, 2016*). We have re-analysed a data-set of 478 individuals genotyped at 14 microsatellite loci from *Vignaud et al. (2014)*. The second example is the leatherback turtle (*Dermochelys coriacea*), the most widely distributed sea turtle found from tropical to sub-polar waters. The global population has been reduced in about 40% in the last three generations. As species, the leatherback turtle is classified as vulnerable, mainly because of the Northwest Atlantic population that shows an increase in number of nests. However, other populations are critically endangered (*Wallace, Tiwari & Girondot, 2013*). The data-set re-analysed (215 individuals genotyped at 10 mocrosatellite loci; *Molfetti et al., 2013*) comes from the Northwest Atlantic population. Last, we re-analysed the data from the populations of two colobus monkeys at the Cantanhez National Park in Guinea Bissau (*Minhós et al., 2016*). The Western black-and-white colobus (*Colobus polykomos*, 22 individuals genotyped at 14 loci) and the Temminck's red colobus (*Procolobus badius* ssp. *temminckii*, 23 individuals genotyped at 13 loci) are two sympatric species from the Western African rainforest considered to be vulnerable and endangered respectively (*Oates, Gippoliti & Groves, 2008*; *Galat-Luong et al., 2016*). Data were analysed with the same prior distributions as PODs except for the colobus monkeys datasets, which consist of tetranucleotide markers. Previous evidence suggests that tetranucleotide microsatellite mutations are mainly of only one repeat unit (e.g., *Leopoldino & Pena, 2003*; *Sun et al., 2012*). In order to incorporate this prior knowledge, half of the simulations had $P_{GSM} = 0$ (i.e., a strict stepwise mutation model, SMM) and the other half had the parameter sampled from a uniform distribution in the range $(0, 1)$.

For comparison, demographic history of the four real data sets was also explored using the MIGRAINE software (*Rousset & Leblois, 2016*, http://kimura.univ-montp2.fr/~rousset/Migraine.htm ) under the model of a single panmictic population with an exponential change in population size. To infer model parameters, MIGRAINE uses coalescence-based

importance sampling algorithms under a maximum likelihood framework (*Leblois et al., 2014*) using OnePopVarSize model. In this model, MIGRAINE estimates present and ancestral scaled population sizes ($\theta_0 = 4N_0\mu$ and $\theta_A = 4N_A\mu$) and the scaled time of occurrence of the past change in population size ($D = T/4N$, going backward from sampling time, when the population size change began). The past change in population size is deterministic and modelled using an exponential growth or decline that starts at time $D$. Before time $D$, scaled population size is stable and equal to $\theta_A$. MIGRAINE allows departure from the strict SMM by using a GSM with parameter $P_{GSM}$ for the geometric distribution of mutation sizes. Finally, detection of significant past change in population size is based on the ratio of population size ($\theta_{\text{ratio}} = \theta_0/\theta_A$). $\theta_{\text{ratio}} > 1$ corresponds to a population expansion and $\theta_{\text{ratio}} < 1$ to a bottleneck. If no significant demographic change is obtained, MIGRAINE is run again under a model of stable demography (a single value of $\theta$) for parameter estimation. For the whale shark data set, MIGRAINE analysis was already done in *Vignaud et al. (2014)*. For the leatherback turtle, MIGRAINE was run using 20,000 trees, 200 points at each iteration and a total of 16 iterations. For the colobus monkeys, we considered 2,000 trees, 400 points at each iteration and a total of 8 iterations.

## RESULTS

### Simulations

The general behavior of the method can be described from three example scenarios (contraction with $\theta_0 = 0.4$, $\theta_1 = 40$, $\tau = 0.1$, expansion with $\theta_0 = 40$, $\theta_1 = 0.4$, $\tau = 0.1$ and constant size with $\theta = 40$; mutational model with $P_{GSM} = 0.22$). These examples correspond to intermediate parameter values. Results for all simulations are available in Supplementary Information 1.

The main output of the analysis is the graphical representation (i.e., the skyline plot) of the inferred demographic trajectory. It consists of a plot with three curves, representing the point estimates (median) and 95% HPD intervals of $\theta$ through time. Skyline plots obtained from PODs are congruent with the true underlying demography simulated (Fig. 1), except in the less favorable scenarios with very recent or very small changes in population size (Figs. S2–S8). Although the trajectory of the posterior median of $\theta$ and the true trajectory share the same trend (declining, increasing or constant), they sometimes differ in magnitude or time-scale. This disparity is more prominent for bottleneck scenarios.

For a quantitative criterion to assert demographic change we explored the value of posterior probabilities for constant and variable population size models, similar to the scheme proposed by *Heled & Drummond (2008)*. These probabilities (summarised as Bayes factors in Fig. 2) proved to be useful for distinguishing bottleneck and expansion scenarios from demographic stability, although with lower performance for less favorable scenarios (Figs. S9–S15). Constant size scenarios show no evidence for size change. The power to detect demographic change reduces with smaller sample size and lower number of loci (Fig. 2) because summary statistics are estimated with lower precision.

Changes in population size were co-estimated with the mutational model parameter $P_{GSM}$. Mean absolute error, bias and proportion of replicates for which the true value was

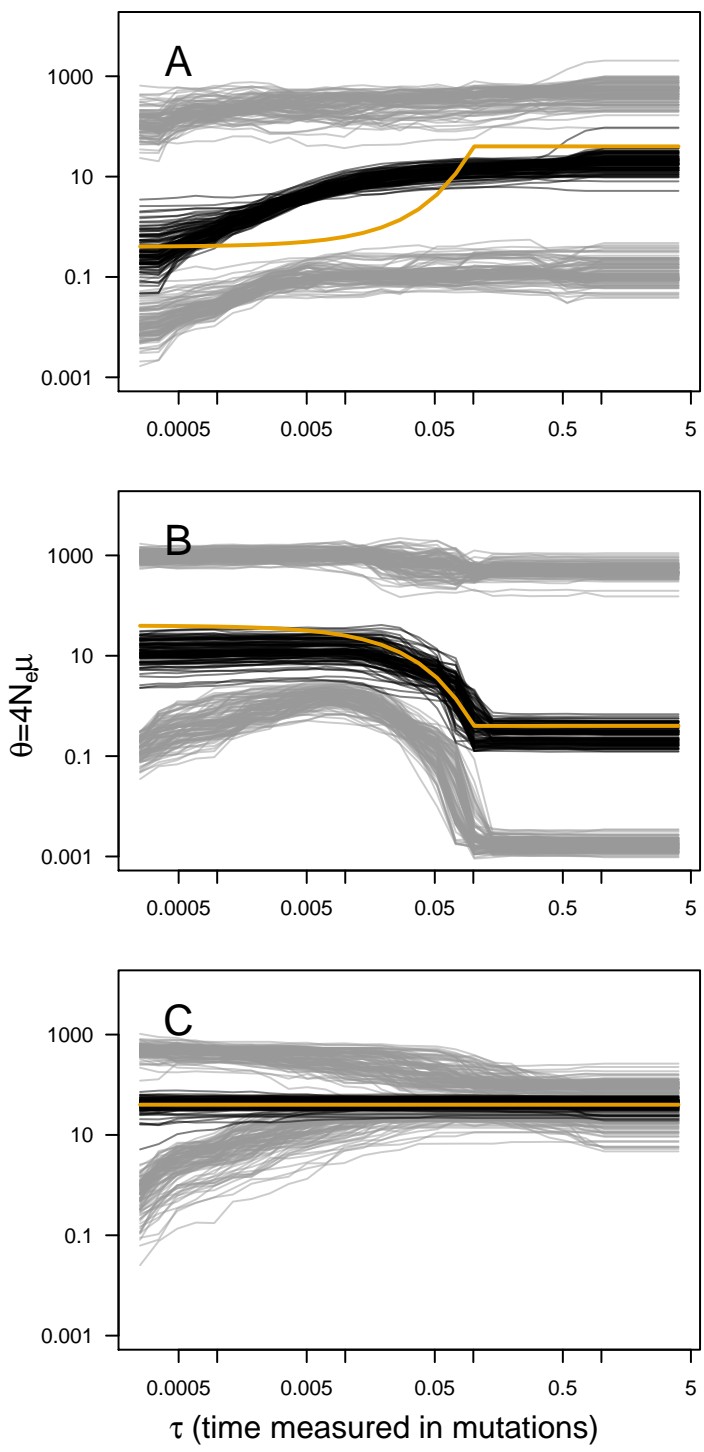

**Figure 1  ABC Skyline plots: simulations.** Superimposed skyline plots (median in black, and 95% HPD interval in grey of the posterior probability distribution for $\theta(t)$) from 100 replicates for example (A) contraction ($\theta_0 = 0.4, \theta_1 = 40, \tau = 0.1$), (B) expansion ($\theta_0 = 40, \theta_1 = 0.4, \tau = 0.1$) and (C) constant size ($\theta = 40$) scenarios with mutational model $P_{GSM} = 0.22$. Simulation of 30 loci sampled at 50 diploid individuals. True demography is shown in orange. Note that present is at $\tau = 0$ (left).

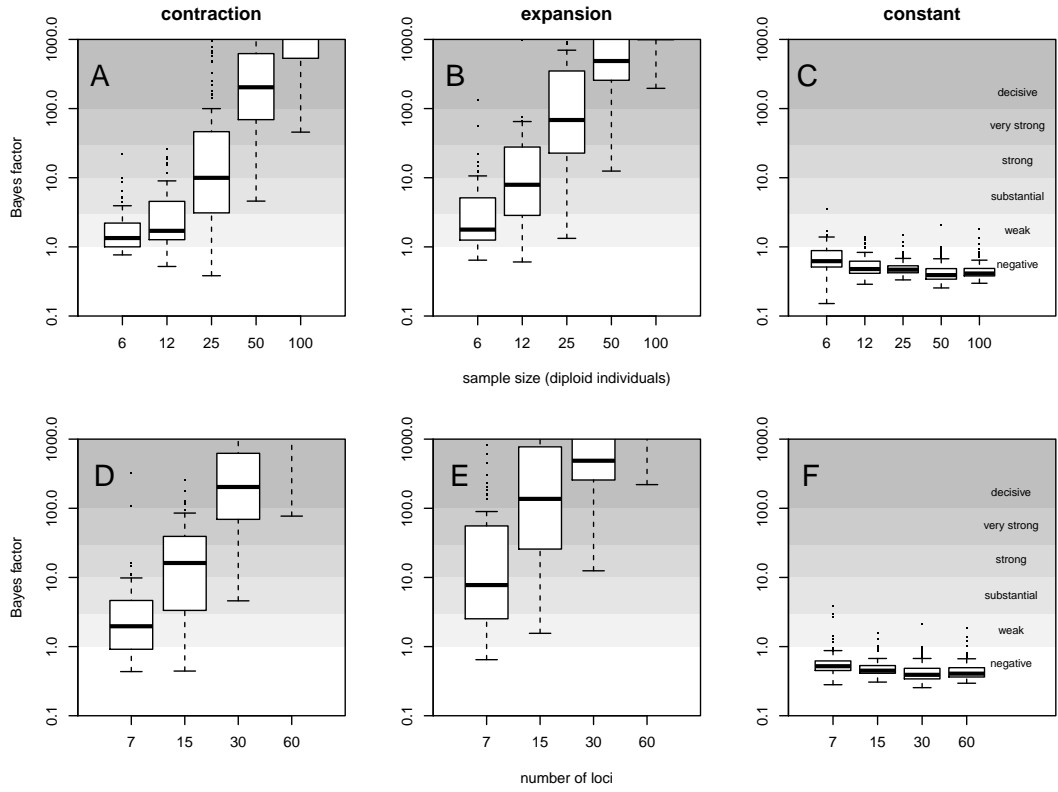

**Figure 2 Evidence for variable population size.** Distribution of Bayes factor values (boxplot) from 100 replicates for example (A, D) contraction ($\theta_0 = 0.4$, $\theta_1 = 40$, $\tau = 0.1$), (B, E) expansion ($\theta_0 = 40$, $\theta_1 = 0.4$, $\tau = 0.1$) and (C, F) constant size ($\theta = 40$) scenarios with mutational model $P_{GSM} = 0.22$. Different sized data sets (number of individuals and loci) are presented, with simulation of 30 loci (A–C) and simulation with 50 diploid individuals (D–F). For reference, *Jeffreys (1961)* scale is given for the evidence against constant size.

**Table 1 Estimation of mutational parameter $P_{GSM}$.**

| Model | $\theta_0$ | $\theta_1$ | $\tau$ | $P_{GSM}$ | MAE | Bias | Out of CI |
|---|---|---|---|---|---|---|---|
| Contraction | 0.4 | 40 | 0.1 | 0.22 | 0.14 | 0.13 | 0.01 |
| Expansion | 40 | 0.4 | 0.1 | 0.22 | 0.05 | −0.04 | 0.05 |
| Constant size | 40 | | | 0.22 | 0.06 | −0.03 | 0.00 |

**Notes.**
MAE, mean absolute error; out of CI, proportion outside credibility interval (95% HPD).
Estimates from 100 replicates.

outside the 95% HPD interval are reported in Table 1 for the three example scenarios and in Table 1 for all simulations. Estimates from expanding and stable populations show a relatively low error and bias and a good coverage of the credibility interval (except in the strict SMM case). However, estimates from declining populations show higher error and bias.

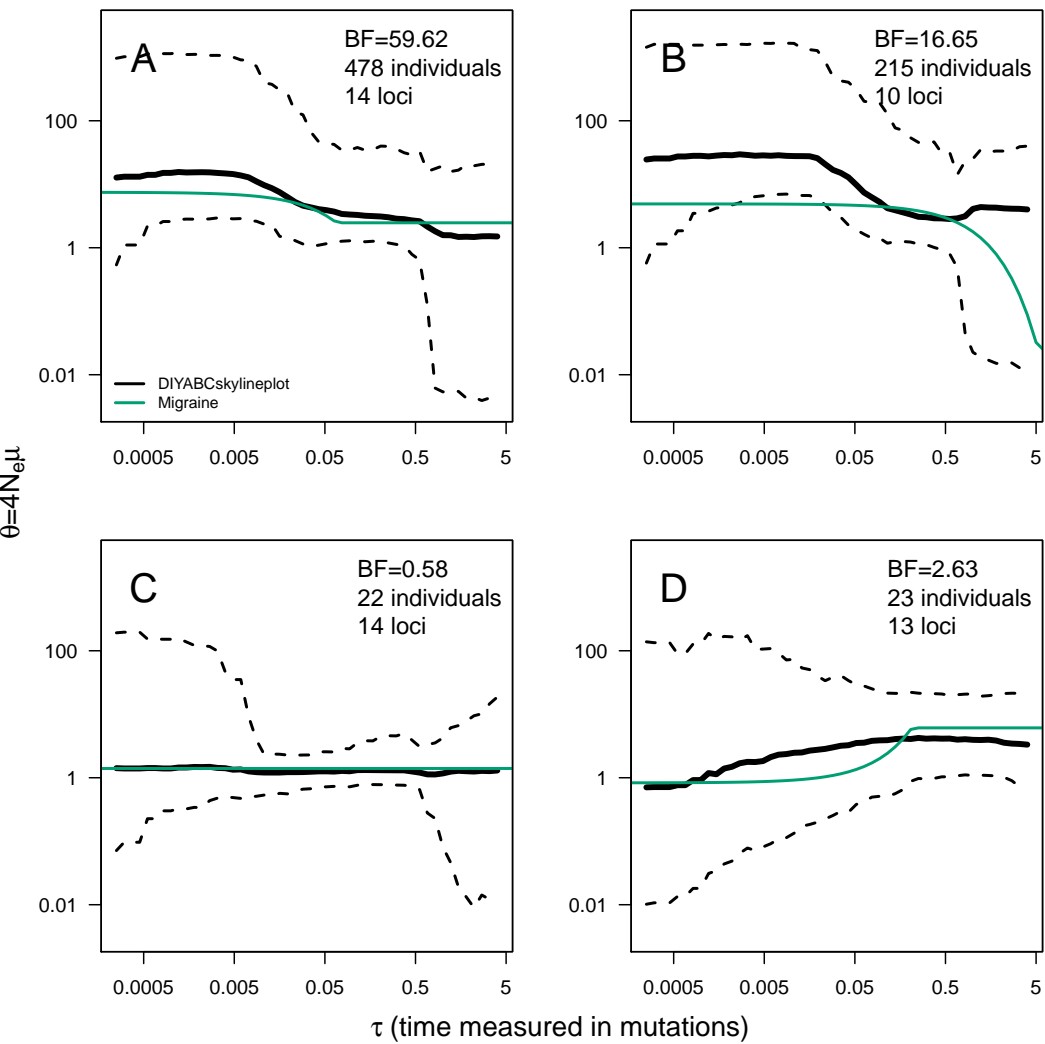

**Figure 3  ABC Skyline plots: real data.** Skyline plots (median in black, and 95% HPD interval in grey of the posterior probability distribution for $\theta(t)$) for whale shark (A), leatherback turtle (B), Western black-and-white colobus (C) and Temminck's red colobus (D). Bayes Factors (BF) are reported for the variable *versus* constant size model. Demographic trajectories based on parameters point estimates from MIGRAINE analysis are shown with a green line for reference. Note that present is $\tau = 0$ (left).

## Real data

The ABC analyses show evidence of population expansion for the whale shark ($BF = 59.62$) and the leatherback turtle ($BF = 16.65$); no evidence for population size changes in the black-and-white colobus ($BF = 0.58$) and some evidence for a bottleneck in the red colobus ($BF = 2.63$). Respective skyline plots reflect such trends (Fig. 3). Results from MIGRAINE support the same trends, with $\theta_{\text{ratio}}$ significantly larger than one for the whale shark and the leatherback turtle, significantly smaller than one for the red colobus and not significantly different from one for the black-and-white colobus (Table S3). Scaled population size estimates through time are also in agreement, except for the leatherback turtle, where the MIGRAINE result suggests a more ancestral expansion of much greater magnitude.

Regarding the mutational model, a large proportion of multi-step mutations seems to be present in all datasets, with $P_{GSM}$ estimates: $\hat{P}_{GSM} = 0.55$ (95% HPD = 0.46–0.62) for the whale shark; $\hat{P}_{GSM} = 0.50$ (95% HPD = 0.38–0.60) for the leatherback turtle; $\hat{P}_{GSM} = 0.43$ (95% HPD = $4.05 \times 10^{-3}$–0.53) for the black-and-white colobus; and $\hat{P}_{GSM} = 0.18$ (95% HPD = 0.02–0.75) for the red colobus (see also Fig. S16). Although very small values of $P_{GSM}$ are included in the credibility interval from the colobus analyses, the GSM is favoured over the SMM when an ABC model choice analysis is performed (BF = 57.50 for the black-and-white colobus and BF = 10.01 for the red colobus). These results are congruent with estimates of $P_{GSM}$ by MIGRAINE (Table S3).

## DISCUSSION

The ability of the ABC skyline plot to detect changes in population size varies largely across the different scenarios evaluated. The evidence for demographic change was often strong (even very strong) in declining and expanding populations. However, demographic changes of small magnitude and close to the present were the hardest to detect. Recent or small magnitude events leave a weak signal in the genetic data and are also hard to identify for alternative methods (see *Girod et al., 2011*; *Leblois et al., 2014*; *Nikolic & Chevalet, 2014*). In any case, the method is conservative, since most analyses of stable populations yielded negative or little evidence for demographic change.

The main appeal of skyline plots is to depict demographic trajectories not bounded by a mathematical function, thus potentially reflecting more realistically the demography of natural populations. However, our results show that plotted trajectories only loosely reflect the true demography, particularly for contracting populations. The match between the true and inferred demographic trajectory was good for constant size populations and for some expanding populations. Ancestral and current population sizes (the extremes of the skyline plot) were also retrieved accurately for favourable scenarios. Nevertheless, the shape of the curve representing the transition between population sizes was a poor representation of the true demographic trajectory in many cases. While this conclusion is specific for the implementation presented in this work, it calls to caution for the interpretation of results from other methods yielding smooth skyline plots (e.g., *Heled & Drummond, 2008*; *Gill et al., 2013*; *Nikolic & Chevalet, 2014*). The key for a smooth skyline plot is the prior on the effective-size autocorrelation through time. The demographic history consists of several demographic periods. Within each period the effective size at consecutive generations is correlated through some mathematical function (often a constant). Between consecutive periods, population size can be independent (our approach) or correlated by different sets of priors. *Drummond et al. (2005)* proposed using an exponential prior for the effective size ($\theta_i$) at period $i$ with mean equal to the previous period effective size ($\theta_{i-1}$). In the Bayesian skyride and skygrid (*Minin, Bloomquist & Suchard, 2008*; *Gill et al., 2013*) the correlation of effective size through time is modelled with a Gaussian Markov random field that penalizes differences in effective size across periods in function of the temporal distance among them. A superficial comparison with the VarEff method (*Nikolic & Chevalet, 2014*) and the extended Bayesian skyline plot (*Heled & Drummond, 2008*) seems to indicate that their inferences suffer from problems of performance (see Fig. S19).

Bottlenecked populations, which show the greatest discrepancy between the skyline plot and the true demographic curve, are also the scenarios for which the mutational parameter $P_{GSM}$ was inferred with largest bias. Similar patterns of summary statistics are produced with large $P_{GSM}$ values and with a bottleneck (e.g., large allele size variance, see Table S2), which make accurate joint inference of demography and mutational models difficult. This difficulty of distinguishing between scenarios with frequent multi-step mutations and contracting populations also explains the low power to detect some bottleneck cases, such as those with large $P_{GSM}$ values and strong declines in population size (see Fig. S11). A negative effect on demographic inference due to mutational model misspecification has been also reported for alternative methods (see *Girod et al., 2011*; *Leblois et al., 2014*; *Nikolic & Chevalet, 2014*).

Globally, our results highlight the interest of using complementary data and inference methods. In the four real-data populations, their demographies have been previously studied in the original publications. In addition to the MIGRAINE analysis of microsatellite data, *Vignaud et al. (2014)* inferred a population expansion for the whale shark by using Bayesian skyline plot analysis on mitochondrial DNA sequence data, corroborating the signal of expansion for this species. In the case of the leatherback turtle, the previous analyses were less conclusive (*Molfetti et al., 2013*). An extended Bayesian skyline plot on microsatellite data suggested an expansion, but it was not significant, and the skyline plot on mitochondrial DNA data did not show any demographic change. In contrast, analysis of microsatellite data with MSVAR (a coalescent sampler approach, *Beaumont, 1999*; *Storz & Beaumont, 2002*) suggested a strong population decline. However, it must be noted that MSVAR assumes a strict SMM, which can lead to biases in the demographic estimates when microsatellite mutations include a substantial proportion of multi-step changes (*Girod et al., 2011*; *Faurby & Pertoldi, 2012*). Our estimates of the $P_{GSM}$ parameter and the two-phase model used in BEAST suggest a strong departure from the SMM and lead us to favour the hypothesis of population expansion. Finally, the original analysis of the two colobus species found significant evidence of population decline for both of them (*Minhós et al., 2016*). Again, this evidence was obtained from MSVAR and the extended Bayesian skyline plot implemented in BEAST assuming a SMM. Despite the prior results suggesting that tetranucleotide microsatellite mutations add or remove a single repeat, our analyses (ABC skyline plot and MIGRAINE) rejected the SMM for the black-and-white colobus. This explains the difference between their results and our demographic inference, which supports a constant size for this population.

Results from demographic inferences have been reported in the form of the scaled parameters $\theta$ and $\tau$ throughout this work. This is because rescaling to natural parameters (effective population size in number of individuals and time in number of generations or years) requires independent knowledge of mutation rates, which is unavailable for most species (including our four study cases). If such knowledge exists, a prior can be used in DIYABC to incorporate this information in the analysis and make inferences on natural scale parameters. Otherwise, we advocate reporting coalescent scaled parameters as results of the analysis. This allows the discussion of the result considering different mutation rates

and reinterpretation of results if information on mutation rates is obtained in the future for the focal species.

A common problem for the inference of population size changes is the presence of population structure or gene flow. Most methods aiming to detect population size change often assume the analysis of a single, independent population, but violation of this assumption usually leads to false detection of bottlenecks (e.g., *Heller, Chikhi & Siegismund, 2013*; *Nikolic & Chevalet, 2014*, for skyline plot approaches). We expect the same effect in the implementation of the skyline plot analysis we present here. However, distinguishing between population structure and population decline in the ABC framework is possible under some circumstances with the appropriate summary statistics (*Peter, Wegmann & Excoffier, 2010*) that can be included in future implementations of the ABC skyline plot.

Indeed, the ease of incorporating new summary statistics and models is of prime interest for implementing the skyline plot in the ABC framework. Multiple samples of the same population at different times (as in experimental or monitored populations and ancient DNA studies) can easily be simulated allowing for better estimates of the effective population size (*Waples, 1989*; *Navascués, Depaulis & Emerson, 2010*). Models with multiple populations can also be simulated and skyline plots for each of the populations estimated. Extensions to other molecular markers will be straightforward to develop and already exist for genomic data (e.g., *Boitard et al., 2016*). Finally, other demographic parameters, such as the migration rate (*Pool & Nielsen, 2009*), could be subject to variation with time and they, too, could be inferred with a similar scheme. To sum up, there is potential to develop this approach in different directions, to address new questions in future research.

In this work we presented a detailed description of how to compute an approximate-Bayesian-computation skyline plot and assessed its performance on stable and changing simulated populations characterized with microsatellite markers. Its power to detect the signal of demographic change is similar to alternative methods. However, its potential ability to depict the demography of natural populations more realistically must not be overrated. Still it offers an analysis complementary to other methods and there is great potential to develop it to cover other models and types of genetic data.

## ACKNOWLEDGEMENTS

We thank Jean-Marie Cornuet and Renaud Vitalis for helpful discussion on this project and Alexandre Dehne-Garcia for his technical support. We thank Benoît de Thoisy and Tania Minhós for kindly sharing their data on the leatherback turtle and colobus monkeys, respectively. Suggestions from two anonymous reviewers helped to improve this work and Ruth Hufbauer kindly revised and improved the English of the manuscript. All analyses were performed on the INRA MIGALE (http://migale.jouy.inra.fr), GENOTOUL (Toulouse Midi-Pyrénées) and CBGP HPC bioinformatics platforms.

### Funding

This work was supported by the the INRA (Jeune Equipe IGGiPop) and the Investissements d'Avenir (Institut de Biologie Computationnelle-IBC). The funders had no role in study design, data collection and analysis, decision to publish, or preparation of the manuscript.

### Grant Disclosures

The following grant information was disclosed by the authors:
INRA.
Investissements d'Avenir.

### Competing Interests

The authors declare there are no competing interests.

### Author Contributions

- Miguel Navascués conceived and designed the experiments, performed the experiments, analyzed the data, contributed reagents/materials/analysis tools, wrote the paper, prepared figures and/or tables, reviewed drafts of the paper.
- Raphaël Leblois analyzed the data, wrote the paper, reviewed drafts of the paper.
- Concetta Burgarella conceived and designed the experiments, analyzed the data, reviewed drafts of the paper.

### Data Availability

Code for DIYABCskylineplot is available on ZENODO (doi: 10.5281/zenodo.267182) and on GitHub (http://github.com/mnavascues/DIYABCskylineplot) including code to automatically simulate pseudo-data. Data from whale sharks is available at DRYAD database (doi: 10.5061/dryad.489s0 ).

### Supplemental Information

Supplemental information for this article can be found online at http://dx.doi.org/10.7717/peerj.3530#supplemental-information.

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
