# Peer review of "Demographic inference through approximate-Bayesian-computation skyline plots"

_PeerJ, doi:10.7717/peerj.3530_

## Round 0.1 · original submission · Major Revisions

Dear Authors,

I have now received two reviews; both recommend major revision. I am also recommend major revision.

Both reviewers provided and suggestions on how to improve the article, and I would like you to follow these suggestions. They will improve the MS significantly. When you send in a revision, if you choose to resubmit this MS, please provide a detailed response to the reviewer’s comments as well.

In my own opinion, as presented, the article has a number of problems and weaknesses, many of them addressed by the reviewers, but here are the principal points I am concerned with.
1) The MS needs to be edited for style and orthography. As written, there are many statements that are awkward, and at times even unclear on first reading. Please have the MS revised by a native English speaker.

2) The MS does not provide sufficient detail when the skyline plot ABC proposed in this MS is compared to other methods. Thus I would like to see: a) better explanation of the assumptions and the workings of other demographic methods (Skyline and Skyride methods, MSVar), b) reanalyses of the data using these methods which will then make it easier for the reader to visualize and understand the differences of the results generated by all the methods.

3) The potential limitations and sensitivity of the skyline plot ABC method should be addressed (see comments of the first reviewer).

Otherwise, I think this MS and the presented method will make a solid contribution to demographic analyses of microsatellite data. In the near future as NGS data will become more prevalent, and I hope to see this method extended to SNP or short read sequence data.

Sincerely,

Tomas Hrbek

Reviewer 1 ·

Basic reporting

This manuscript was clear in its presentation. I offer the following grammatical suggestions.
22: type => types
31: describe => describes [The subject ‘model’ is singular]
61: delete ‘to make’
79: build => built
100: delete ‘be’
102: number => numbers
181: delete ‘and their results will be presented in the main text.’ [This is understood]
185: in => of estimate => estimates [the plot and CIs are lines that represents numerous ‘point’ estimates that have been integrated into a smooth line]
186: interval => intervals
208: higher => larger lower => smaller
209: different than => different from [‘than’ is American slang]
216: low => small
226: shows to be => is
228: Skyline-plot main appeal is => The main appeal of skyline plots is
228: bounded to a => bounded by a
230: trajectories loosely => trajectories only loosely those of => for
234: them => population sizes [the antecedent of this pronoun is unclear]
241: worth to mention that => worth mentioning that
247: value => values
257: delete ‘found’
258: On contrast, => In contrast,
272: aiming the detection of population => aiming to detect population
273: violating => violation conduce => leads
279: is the prime => is of prime
Figure 1: [Broaden the orange line to make it more visible. Also the lower HPD traces in light grey are difficult to see. These could be the same shade of grey as the upper HPD traces.]
Figure 2: Bayes factor distribution => Distribution of Bayes factors
Supplementary Methods
S1.3
Line 8: implement => implements
Figures S2, S3, S4, S5, S6, S7, S8, S17: Please darken the lower HPD traces. They are difficult to see.

Experimental design

The use of skyline plots generated by coalescence analysis of DNA data to infer a population’s past demographic history has been enthusiastically embraced by population geneticists and phylogeographers. Two problems, however, have been limitation of methods to analyse only sequences, to the exclusion of other kinds of DNA data and of the large computation times needed to generate a skyline plot, especially of very large datasets. This study further explores the development of analysing microsatellite datasets and outlines a strategy to use ABC methods to speed analysis. With the development of larger and larger genomic data sets that do not consist of sequences, this study represents a step forward in making skyline plot analysis more widely available to molecular biologists.
The authors explored the parameter space of several variables including the nature of mutation in microsatellite, the timing of a past demographic event, ancestral and contemporary population sizes and the three kinds of demographic history through simulations. Estimation of mutation rate to calibrate a molecular clock can also affect a skyline plot, but this problem was avoided by using tau in the skyline plots, rather than attempting to place demographic events in absolute time. Nevertheless, the demographic events in the skyline plots of ‘real data’ are described as ‘recent’. How was this concluded in the absence of a temporal calibration. The literature is full of demographic expansions that supposedly occurred tens of thousands or even hundreds of thousands of years ago.
Lacking in this experimental design was the exploration of the effect of sample size, which is expected to have some influence on the shape of a skyline plot, especially for microsatellites because of the prevalence of high allelic diversity and an abundance of low frequency alleles. The flat skyline curves for the species of colobus may in fact be due to small samples sizes (n = 22 and 23) and may not reflect true demographic histories.
Another problem not addressed in the manuscript are assumptions in the model that produced the simulated datasets and in the model generated the skyline plots from these datasets. Most DNA simulation programs and BEAST, for example, are based on the Wright-Fisher model of allelic change, which assumes that only bifurcations occur at a genealogical coalescence. Theoretical work (several articles by Wakely, Eldon and others), however, show that the genealogies of many species commonly contain multiple-mergers that greatly influences the shape of a skyline plot. The excess of low-frequency alleles that are widely interpreted to reflect demographic expansions, or in some cases natural selection, may in fact reflect reproductive skew in some species.

Validity of the findings

The results of this study appear to be sound over the parameter space that was included in the analyses and will help in the analysis of non-sequence data such as microsatellites. Additionally, the use of ABC approaches for generating skyline plots will help in the analysis of larger datasets, which are becoming more and more common. The extension of skyline plot estimation to non-sequence data is also step forward.

Reviewer 2 ·

Basic reporting

This manuscript presents a potentially useful method for estimating changes in population size through time, using microsatellite data. The authors illustrate the performance of this method on simulated data and on 4 real data sets. The method performs well in some situations but poorly in others.

There are numerous minor grammatical errors throughout the manuscript, including several run-on sentences. The manuscript needs careful editing/proofreading.
- Line 55. “underlying”.
- Line 62. “The incorporation of microsatellites to the software BEAST allowed to make skyline plot inference for this type of data” should be rephrased to something like “The addition of microsatellite mutation models to the software BEAST made it possible to infer skyline plots from this type of data”.
- Line 68. It is not clear what the authors mean with “However, if a large proportion of the genome has been sequenced, the studied polymorphism are not independent”. Is this referring to linkage disequilibrium?

- The authors could also point out that adding ancient genetic data can help to resolve past changes in population size (e.g. Mourier et al. 2012 in Mol Biol Evol).

- Line 52. The authors should briefly describe the original work on (non-Bayesian) skyline plots by Pybus and colleagues.

Experimental design

- To allow readers to gain a better assessment of the proposed method, the authors should provide more detail about the comparison of the ABC skyline method with other methods. In the present manuscript, these results are presented in Supplementary Figure S19 and are only mentioned very briefly. I suggest moving this figure to the main text and providing a more detailed discussion of the performance of the different methods. I would also like to see the other methods being used to analyse the real data (not just MIGRAINE). The authors have mentioned that some of the data sets were analysed using BEAST, but it would be useful to replicate those analyses for the present study and to present the plots of the inferences from the different methods.

- I would also like to see some exploration of how the method performs with different sizes of data sets. For example, showing the resolution of the demographic history for different numbers of microsatellite loci would provide a useful guide for users of the method.

- One important aspect of the analysis that is missing from the manuscript is the choice of microsatellite mutation rate. This is important for accurate rescaling of the skyline plot. How is uncertainty in the mutation rate incorporated? I am surprised that this issue is not even discussed, given that knowledge of the mutation rate is essential for addressing the goals described at the start of the Introduction (e.g. identifying the factors that drove past changes in population size).

- Line 236. The authors should briefly compare the Bayesian skyline and skyride methods, which make different assumptions about correlated population sizes through time.

Validity of the findings

See comments above.

Additional comments

No further comments.

---

## Round 0.2 · Minor Revisions

Dear Authors,

I have now received two re-reviews, both recommending minor revision. Both reviewers still have concerns that they would like you to address, and I would like you to address them as well.

Reviewer 1.
I would like you to address the issue of sample size. Although it may seem obvious, please mention that the detection of demographic expansion is predicated on the detection of low frequency alleles, and thus small sample sizes may result in false negatives. Or rather, a given sample size may be too small to detect a particular demographic expansion, i.e. one needs larger sample sizes to detect smaller magnitude demographic events.

Reviewer 2.
As per reviewer’s 2 suggestions, please make the distinction between skyline and skyride plots and methods.

Otherwise, I will be happy to recommend acceptance once you address the reviewer’s comments.

Sincerely,

Tomas Hrbek

Reviewer 1 ·

Basic reporting

1. Basic reporting: The revision presents the authors' argument in a logical order, and problems with English grammar have been well addressed. However, the title might be more informative by including the phrase 'microsatellite DNA'. While much of the discussion can also apply to DNA sequences, the simulations and the examples were based on microsatellites. This does not appear in the title.

Experimental design

2. Experimental design: The design remains essentially the same. The authors adequately argue against including some additional analyses suggest by the two reviewers. The simulations appear to adequately support the core of the authors' conclusions.

Validity of the findings

3. Validity of findings: The authors convincingly demonstrate that an ABC approach to estimating skyline plots has merit.

Additional comments

4. General comments: One difficulty in using MCMC Bayesian methods for estimating BSPs has been the amount of computer time needed for a run. Most empirical phylogeographers do not have access to supercomputers so the development of methods to improve the analysis of especially large datasets is welcome. However, I am still concerned by the small sample sizes used in some of the simulations. The earmark of a demographic expansion is the accumulation of low-frequency mutations, which cannot be sampled with small sample sizes. Small samples sizes reduce the power of a BSP analysis to detect a population expansion and can lead to an increase in type II error. I realize that some species are difficult to sample, but the use of small sample sizes can lead to erroneous conclusions, which may be detrimental in a conservation context.

Nevertheless, I think the manuscript is now suitable for publication.

Reviewer 2 ·

Basic reporting

The authors have addressed most of the points raised in my previous review, and the revised manuscript is now much clearer to read.

In my previous review I suggested that the authors should briefly compare the Bayesian skyline and skyride methods, but their response was that these deal with single-locus data. However, the Bayesian skygrid is a multi-locus generalization of the Bayesian skyride. A brief discussion of how these methods differ would be useful, especially because the authors mention the issue of smooth demographic plots (line 250).

- Lines 160-165. Please provide more background information about the four real data sets here.

- Lines 249-251. This caution sounds too strong. In some cases the change in population size might actually be quite smooth. The smoothness of the plot also depends on the prior on population-size autocorrelation through time.

Minor comments
- Line 16. Replace “skyline plot” with “skyline plots”.
- Line 58. Replace “consists in” with “consists of”.
- Line 221. Replace “no significantly” with “not significantly”.
- Line 223. Replace “where MIGRAINE result” with “where the MIGRAINE result”.
- Line 267. Please rephrase “These results highlight the interest of using”.
- Line 279. Replace “analysis on” with “analysis of”.

Experimental design

I have no major concerns about the analyses.

Validity of the findings

No comment.

---

## Round 0.3 · accepted · Accept

Dear Authors,

Thank you for submitting your manuscript. I am happy with the revisions, and how you addressed the reviewers’ comments, and therefore I am recommending acceptance.

Congratulations on a job well done.

Sincerely,

Tomas Hrbek